# Key Role of Precursor Nature in Phase Composition of Supported Molybdenum Carbides and Nitrides

**DOI:** 10.3390/ma12030415

**Published:** 2019-01-29

**Authors:** Zdeněk Tišler, Romana Velvarská, Lenka Skuhrovcová, Lenka Pelíšková, Uliana Akhmetzyanova

**Affiliations:** Unipetrol Centre for Research and Education, a.s, Areál Chempark 2838, Záluží 1, 436 70 Litvínov, Czech Republic; Zdenek.Tisler@unicre.cz (Z.T.); Romana.Velvarska@unicre.cz (R.V.); Lenka.Skuhrovcova@unicre.cz (L.S.); Lenka.Peliskova@unicre.cz (L.P.)

**Keywords:** Molybdenum nitride, molybdenum carbide, hexamethylenetetramine

## Abstract

In this work, we studied the effect of molybdenum precursors and the synthesis conditions on the final phase composition of bulk and supported molybdenum carbides and nitrides. Ammonium heptamolybdate, its mixture with hexamethylenetetramine, and their complex were used as the precursors at different temperatures. It was investigated that the synthesis of the target molybdenum nitrides strongly depended on the structure of the precursor and temperature conditions, while the synthesis of carbide samples always led to the target phase composition. Unlike the carbide samples, where the α-Mo_2_C phase was predominant, the mixture of β-Mo_2_N, MoO_2_ with a small amount of metal molybdenum was generally formed during the nitridation. All supported samples showed a very good dispersion of the carbide or nitride phases.

## 1. Introduction

There are three types of bonding between the transition metal and carbon or nitrogen atoms: metal bonding (metal-metal), covalent bonding (metal and non-metal), and ionic bonding (charge between metal and non-metal) [1]. The special crystal structure of transition metal carbides and nitrides is created by inserting carbon or nitrogen into the metal–metal bond is what makes its distance longer than the original. This special bond has exclusive electronic properties, which provide catalytic activity similar to the platinum group metals (Pt, Pd, Ru, etc.) in various reactions [2]. Molybdenum carbides exist in three basic forms: a face-centered cubic (fcc, α-MoC_1-x_), a hexagonal closed packed (hcp, β-Mo_2_C), and a simple hexagonal (hex, MoC) structure, while molybdenum nitrides mainly have a cubic structure (fcc, γ-Mo_2_N) [3].

Molybdenum carbides are widely used as catalysts due to their activity in many reactions, particularly in the water gas shift reaction, deoxygenation, denitrification, desulfurization, oxidation, partial oxidation, hydrotreating (HDS, HDO, HDN), dehydrogenation, isomerization, hydrogenolysis, hydrodemetallization, and methane reforming [4]. Molybdenum nitrides possess a series of unique and superior catalytic properties for HDS, HDO, HDN [4,5], and electrochemical catalysis [6,7]. Possible difficulties in the application of these types of catalysts that may appear are most often related to obtaining those materials with high specific surface areas (usually less than 10 m^2^/g) or high porosity. However, these parameters may be varied by changing of synthesis conditions.

High surface area Mo_2_N and Mo_2_C are synthesized by various methods [8,9]: (a) Direct reaction between metal and non-metal; (b) reaction of metal oxide in the presence of solid carbon; (c) reaction of metal or compounds with gas phase reagent; (d) temperature-programmed methods; (e) reaction between metal oxide vapor and solid carbon under vacuum; (f) pyrolysis of an organometallic complex under H_2_, and other methods where thin carbide films are obtained by chemical vapor deposition [10], plasma method, or electrochemical reduction [11,12,13].

The oldest method for synthesis of molybdenum carbides is the direct reaction of metals or metal oxides with elemental carbon in a vacuum or reducing atmosphere of Н_2_ at 1200–2000 °C, and a self-propagating high-temperature synthesis. The indicated methods produce only bulk metal carbides with low specific surface areas [14].

Using carbon black (CB) as a carburizing agent is a completely new technique for molybdenum carbide preparation. In this synthesis, a mixture of ammonium heptamolybdate (AHM) solution with a CB suspension in acetone is heated up to 350 °C in the air. Then, the obtained МoО_3_/СВ precursor is carbidized in a gas mixture of СН_4_/Н_2_ [15,16,17]. Recent publications reported an application of organic compounds (ethylene glycol, glucose, sucrose, polymers) as a source of carbon [18,19,20,21]. The organic material is dissolved in water and mixed with molybdic acid or AHM to obtain cubic molybdenum carbides with small crystal domain sizes at a high temperature. In the past decade, a synthesis of metal carbides with the use of mechanical activation was also published [22,23,24,25]. For the synthesis of molybdenum carbides using mechanical activation, CB was impregnated with an aqueous solution of AHM by the incipient wetness method. Synthesized materials showed a good mechanical resistance and relatively high specific surface area (about 125 m^2^/g).

Preparation of molybdenum nitrides is based mainly on ammonia reduction of molybdenum-containing precursors. Thereby, a high surface area Mo_2_N (116 m^2^/g) was prepared through nitridation of MgMoO_4_ in a flow of N_2_/H_2_ gas mixture at 800 °C [26]. However, the general method for synthesis of molybdenum carbides and nitrides is temperature-programmed reduction (TPR), where carburization/nitridation is carried out by a reaction between molybdenum oxide (MoO_3_ or MoO_2_) and a gas mixture, such as H_2_ with carbon- (CH_4_, C_2_H_6_, C_3_H_8_, toluene, etc.) or nitrogen-containing gases (NH_3_, N_2_).

Different synthesis conditions can also change molybdenum structures, as in the case of (i) a hexagonal closed packed form of molybdenum carbide (hcp; β-Mo_2_C) using typical temperature ramps [27,28,29,30], (ii) cubic molybdenum carbide phases (fcc; α-MoC_1-x_) through a special preparation and lower temperatures [31,32,33], or (iii) a cubic molybdenum nitride phase (γ-Mo_2_N) [9,26]. These methods allow produce not only metal carbides and nitrides as powders or particles, but also supported species [34,35,36] with a high specific surface area. Molybdenum carbides with a high specific surface area are usually synthesized using AHM as a precursor [37]. AHM is dissolved in water and supplemented with HCl to precipitate molybdic acid, which is then dried and held in hydrogen flowing at a high temperature (1500 °C). By this method, very well crystallized and phase pure catalysts characterized by a small specific surface area are produced. Enlargement of the surface area can be obtained by a single-step carbidization of the precursor (AHM or MoO_3_). When a mixture of CH_4_/H_2_ is used as the reaction gas and temperature-programmed heating is up to 500–900 °C for 3 to 6 h [15,38,39,40,41], a hexagonal phase of molybdenum carbide is formed from AHM, while oxides can be transformed into other carbide structures using toluene as the carburizing agent [30,31,42]. The selected ramps and temperatures allow the formation of the cubic phase instead of hexagonal at low temperatures, without the use of any structure stabilizing agents.

In a number of studies, molybdenum nitride and carbide have been prepared by a thermal treatment of single-source precursors. It is a simple and effective method to obtain Mo_2_C or Mo_2_N using only hexamethylenetetramine (HMT) as a reducing agent and a carbon or nitrogen source [8,43,44,45,46]. Afanasiev [43] studied the thermal decomposition of a precursor synthesized by reacting HMT with AHM. The resultant HMT-AHM precursor complex [(HMT)_2_(NH_4_)_4_ Mo_7_O_24_•2H_2_O] was decomposed under an argon atmosphere in the temperature range 550–800 °C. Using this method, crystalline Mo_2_N was synthesized at the temperature higher than 650 °C, while Wang et al. [45] obtained β-Mo_2_C by mixing HMT with HMT-AHM at the 7:1 molar ratio and at 700 °C in an argon atmosphere.

The main purpose of this study is to gain an understanding of the influence of various preparation methods of a precursor on its final physical, chemical, and textural characteristics. Wang et al. [44] have already presented two basic preparation methods. In the first, authors followed the method reported by Afanasiev [43] and obtained Mo_2_C, while in the second, they used mechanically mixed AHM and HMT with a molar ratio of 1:4 with obtaining metallic Mo and MoO_2_. The researchers also described an impact of different parameters in hydrogen thermal treatment preparation of HMT and AHM, namely, AHM:HMT molar ratios, temperature, or heating rates [44]. They reported that pure Mo_2_C was obtained when the molar ratio of precursors AHM and HMT reached 1:4, and nitride, carbide, and carbonitride composite materials were obtained when the molar ratio was 1:2. They showed that Mo_2_N became a major product phase at a lower temperature (500 °C), and for Mo_2_C a higher temperature (650 °C) was needed.

## 2. Materials and Methods

### 2.1. Materials

Analytical grade ammonium heptamolybdate (AHM), hexamethylenetetramine (HMT), and ammonia solution (25 wt%) were supplied by Lach-Ner s.r.o. (Neratovice, Czech Republic). The γ-Al_2_O_3_ spheres (diameter 2.5 mm) were purchased from Sasol (Hamburg, Germany), and extrudates of zeolite Beta, TiO_2_, and ZrO_2_ were purchased from Euro Support Manufacturing Czechia, s.r.o. (Litvínov, Czech Republic). Mesoporous silica SBA-15 (powder) and pellets of alkali-activated zeolite foam (AZF) with diameter 5 mm were synthesized in the laboratory.

### 2.2. Preparation of Precursors

Based on the method reported by Afanasiev [43], the initial HMT-AHM precursor complex was synthesized using 50 g of AHM and 86 g of HMT dissolved in 300 mL and 400 mL of distilled water, respectively. The solutions were mixed out together and left for 48 h at 3 °C. The sedimented crystals were separated using a paper filter, rinsed with demineralized water, and dried at room temperature for 3 days. The resulted material was named HMT-AHM 7.5×. The samples of HMT-AHM 5× and HMT-AHM 10× with lesser and greater amounts of HMT, correspondingly, were synthesized with the same method.

Mechanically mixed samples of the HMT+AHM (2:1) and HMT+AHM (8:1) precursors were prepared by grinding in a mortar and pestle for 10 min of HMT with AHM in the corresponding molar ratios 2:1 and 8:1. HMT+AHM-S (1:1) and HMT+AHM-S (2:1) precursors with molar ratios 1:1 and 2:1 were synthesized by evaporation of the ammonia solution of HMT with AHM. MoO_3_ was prepared by calcination of AHM at 450 °C for 6 h.

Precursors for the supported samples were prepared by incipient wetness impregnation of the HMT-AHM 7.5× ammonia solution on the supports Al_2_O_3_, TiO_2_, ZrO_2_, SBA, BEA, and AZF. Al_2_O_3,_ impregnated with HMT+AHM-S (2:1), was signed as Al_2_O_3_^#^. To obtain a high content of molybdenum nitride or carbide phase, the impregnation was performed with a saturated solution. The impregnated support from the same batch was used for the carbide and nitride synthesis. In the case of the AZF support after drying at 120 °C for 6 h, the impregnation was repeated once more.

### 2.3. Synthesis of Molybdenum Carbides and Nitrides

The synthesis of the final nitrides and carbides was carried out in a vertical quartz tubular reactor (UniCRE, Litvínov, Czech Republic) with an internal diameter of 27 mm and length of 1 m (Figure 1), heated to the working temperature by a triple-zone electric oven (CLASIC CZ, spol. s.r.o., Řevnice, Czech Republic) that was regulated by a PID (proportional–integral–derivative) controller. Each precursor or the impregnated support was placed in a fritted quartz cuvette and placed in the center of the reactor. Further processing of the precursors was done in several steps:
Heating (10 °C/min) of the precursor in the N_2_ flow (75 cm^3^/min) at 200 °C for 12 h;Heating to the desired temperature (of 10 °C/min) in a working gas flow (75 cm^3^/min). A mixture of 20 vol% H_2_ in N_2_ was used to prepare nitride samples and a mixture of 20 vol% CH_4_ in H_2_ for the carbide preparation. After reaching the desired temperature, the reaction was run for 3 h;Cooling to the room temperature in the working gas flow (75 cm^3^/min);Flushing the reactor with nitrogen (400 cm^3^/min) for 30 min;Passivation in 1 vol% O_2_ in Ar (75 cm^3^/min) for 2 h.

### 2.4. Characterisation

The chemical composition of the supported samples was determined by X-ray fluorescence analysis (XRF) of powder materials using S8 Tiger (Bruker AXS GmbH, Karlsruhe, Germany) with the Rh cathode. The results were interpreted using the Spectra plus software. The non-supported samples were analyzed by the ICP method using ICP-EOS Agilent 725 (Agilent Technologies Inc., Santa Clara, CA, United States). The carbon and nitrogen content was determined by the elemental analysis of the catalyst powder using Flash2000 Elemental Analyzer (Thermo Fisher Scientific S.p.A., Milan, Italy).

The crystallography of all synthesized material catalysts in powder form was analyzed by X-ray diffraction (XRD) analysis using D8 Advance ECO (Bruker AXS GmbH, Karlsruhe, Germany), applying CuKα radiation (λ = 1.5406 Å) with a resolution of 0.02° and a period of 0.5 s. The patterns were collected in the 2 theta range of 5–70° and evaluated by using the DIFFRAC.EVA software (Bruker AXS GmbH, Karlsruhe, Germany) with the Powder Diffraction File database (PDF 4+ 2018, International Centre for Diffraction Data).

The textural properties of the samples were determined by N_2_ physisorption and mercury porosimetry. The specific surface area (BET) was measured by N_2_ adsorption/desorption at 196 °C by using Autosorb iQ (Quantachrome Instruments, Boynton Beach, FL, United States). All the samples were dried under vacuum before the analysis in a glass-cell at 200 °C for 16 h.

The visual appearance was studied by scanning electron microscope (SEM) JSM-7500F (JEOL Ltd., Tokyo, Japan) with a cold cathode-field emission SEM (parameters of measurements: 1 kV, GB high mode). Additional images were obtained by using an optical microscope Jenavert (Carl Zeiss Microscopy GmbH, Jena, Germany) equipped with a Canon EOS 1200D camera (CMOS chip 18 Mpx, Canon, Taiwan). Images with different focusing were folded by the QuickPHOTO CAMERA software (PROMICRA, Prague, Czech Republic).

Physical properties and stability of the precursors and samples were studied by thermogravimetric analysis (TGA) using TGA Discovery series (TA Instruments, New Castle, DE, United States) operating in the temperature range of 40–900 °C (heating 10 °C/min) in the nitrogen flow (20 mL/min, Linde 5.0). A Quadrupole mass detector OmniStar GSD320 (Pfeiffer Vacuum GmbH, Wien, Austria) was used for detection of fragments in SCAN mode with 1450 V voltage of the electron multiplier. Thermal behavior of the samples was analyzed by differential scanning calorimetry (DSC) using Q2000 (TA Instruments, New Castle, DE, United States). Approximately 5–10 mg of a sample was placed into Tzero pierced aluminum pans. An initial temperature was equilibrated at 0 °C, then the samples were cooled down to −50 °C at a rate of 10 °C/min and held for 1 min. After this, the samples were heated up to 450 °C (10 °C/min).

## 3. Results and Discussion

### 3.1. Precursors

Initially, before the synthesis, all of the precursors were characterized by several analytical methods. As seen from Figure 2, X-ray patterns of the precursor structures differ. In the sample HMT+AHM-M (8:1) produced by mechanical mixing of HMT and AHM, the presence of diffraction lines appropriated to AHM and HMT was observed. However, the line corresponding to the formation of the HMT-AHM complex was also present in the sample.

The precursor complexes, synthesized using HMT+AHM-S, were similar, but not completely the same to those reported by Afanasiev. The MoO_3_ sample had typical diffraction patterns corresponding to molybdenum trioxide (molybdite).

The influence of the HMT excess on the HMT-AHM complex prepared by the Afanasiev method was investigated by varying the molar ratio of HMT:AHM in the sequence of 15:1 (sample HMT-AHM 7.5×), 10:1, and 20:1 (samples HMT-AHM 5× and 10×). The TGA results (Figure 3) show that the forming complexes were absolutely identical. Their similarity was confirmed by the same decomposition in terms of mass depletion of the final residues and decomposition rates at the corresponding temperatures.

Decomposition of other precursors differed from the HMT-AHM 7.5× sample. Thus, in the case of HMT+AHM-M (2:1) and HMT+AHM-M (8:1), a conspicuous signal at 200 °C inherent to HMT decomposition was observed. The crystallization of the HMT and AHM ammonia solution with a molar ratio of 1:1 in the sample HMT+AHM-S (1:1) led to the formation of large transparent crystals that decomposed differently than when only tiny crystals were formed in the sample HMT+AHM-S (2:1). In spite of the fact that the TGA curves of the latter and HMT-AHM 7.5× went through the same trajectory and were very similar, a slight difference in their structures was noticeable.

TGA decompositions of AHM and HMT-AHM were determined using a mass spectrometer. Using the literature data [47,48], it was possible to determine the individual decomposition steps. It was found that thermal destruction at temperatures from 100 to 190 °C resulted in a simultaneous release of H_2_O and NH_3_, giving transition polymolybdate phases. At the same time, fragments corresponding to NO, N_2_O, O_2_, and N_2_ were also observed, which may have been caused by the presence of air in the thermogravimetric furnace, which was not hermetically sealed. The HMT-AHM complex differed by the signal of CO_2_ detected in the range of 190–275 °C, which further gradually released up to 700 °C.

DSC results displayed in Figure 4 show that the HMT-AHM complexes prepared with different HMT excesses (HMT-AHM 5×, 7.5× and 10×), as in TGA, had identical profiles. In comparison with the samples HMT+AHM-M (2:1) and HMT+AHM-M (8:1), some differences to the HMT-AHM complex in the DSC signals were recorded.

HMT+AHM-M (2:1) had a sharp endothermic peak, which occurred in the temperature range of 110–130 °C, and the subsequent endothermic peaks at 250–350 °C correspond to AHM. The change in the molar ratio of HMT:AHM in HMT+AHM-M (8:1) sample also changed the DSC curve. Only two distinct peaks were observable on the record: the first endothermic peak at 170.51 °C corresponded to AHM decomposition and the second one at 248.73 °C to HMT sublimation.

According to the DSC records, the complexes obtained by crystallization of the HMT and AHM ammonia solutions differed among themselves and also in comparison with the HMT-AHM complex prepared by the Afanasiev method. Both HMT+AHM-S (1:1) and HMT+AHM-S (2:1), as well as HMT+AHM-M (2:1) complexes, were characterized by the endothermic peak around 120 °C, which was probably related to the sudden formation of another type of a complex. Despite the fact that the HMT-AHM complex had the same peak, but not quite at intensely, we can conclude that the complex is very similar, but not exactly identical to HMT+AHM-S (2:1). This difference also affected the final structure of molybdenum nitrides and carbides.

SEM results (Figure 5) show that the surface morphology of the HMT+AHM-S (2:1) sample was more crystallized than the HMT-AHM complex. The MoO_3_ sample has small molybdite crystals produced by thermal decomposition of AHM.

### 3.2. Non-Supported MoCx

Non-supported (bulk) molybdenum carbides were synthesised using 20 vol% CH_4_ in H_2_ with a flowrate of 75 cm^3^/min. All the prepared samples showed a high carbon content (Table 1), indicating their complete conversion to carbides. Only the sample where the AHM precursor was used and the reaction temperature was 600 °C contained much less carbon. This was confirmed by X-ray diffraction, where the majority phase was monoclinic MoO_2_ and at the same time a minor orthorhombic modification of α-Mo_2_C occurred. The fractions of the carbide phases and their crystallite sizes are shown in Table 1. As seen from the diffraction patterns of the samples synthesised from the AHM and HMT-AHM precursors at different temperatures (Figure 6), both of them provided a mixture of α- and β-Mo_2_C in the ratio of about 2:1 at 700 °C and only α-Mo_2_C at 800 °C. There was no complete conversion to any carbide phase at low temperatures for AHM (Figure 6a), while HMT-AHM (Figure 6b) gave hexagonal β-Mo_2_C already at 600 °C. Other precursors provided a pure α-Mo_2_C phase only at 700 °C (Figure 7).

The prepared carbides showed a relatively wide range of specific surface area values (Table 1). The samples synthesized from HMT+AHM-S (2:1) had an area two times higher than when HMT-AHM was used. The specific surface areas of the other samples were very small or even equal to zero, though they all contained the carbide phase.

The crystallites sizes (Table 1) were relatively equal. These values were mostly in the range from 14.3 to 21.3 nm for the α-Mo_2_C phase. The exception was AHM-800, the crystallite size of wich was 40.2 nm. The size of the β-Mo_2_C crystalline phase was possible to determine only for the sample HMT-AHM-600, where this phase appears separately. The size of β-Mo_2_C crystallites in this sample was 4.9 nm. In the case of AHM-700 and HMT-AHM-700, crystallite sizes could not be measured due to the overlap of corresponding diffraction lines. The crystallites sizes were calculated from the reflection of 39.5° 2 theta for α-Mo_2_C and 37.1° 2 theta for β-Mo_2_C.

The structure of the prepared non-supported molybdenum carbides was analysed using scanning electron and optical microscopes. The samples synthesised from the HMT-AHM and HMT+AHM-S (2:1) complexes showed a well-crystallized structure, while the microcrystalline structure was peculiar to the samples obtained from AHM (Figure 8). Evaluation of the structure using SEM revealed a distinctive spongy (foam) structure of the prepared samples (Figure 9).

### 3.3. Non-Supported MoNx

All the non-supported (bulk) molybdenum nitrides were synthesised using the same conditions as for carbides. A mixture of 20 vol% H_2_ in N_2_ was used as a working gas. The specific surface area of the prepared materials and the nitrogen content were very dependent on the content and type of the nitride phase (Table 2). The maximum was 29 m^2^/g for the sample prepared from HMT+AHM-M (8:1), followed by HMT-AHM-700. Nitrogen content (Table 2) clearly showed that the nitride phase was produced only under certain conditions, while carbides were formed in the case of each precursor and temperature. A distinctive feature in nitride samples was the presence of carbon, indicating the presence of the carbide phase. The highest carbon and nitrogen were typical for the samples produced from HMT-AHM (700, 800 °C) and HMT+AHM-S (2:1) precursors. The presence of carbon can be explained by the incomplete decomposition of HMT, which was present in the structure of the used precursor complex. The minor phase was confirmed as orthorhombic carbide phase α-Mo_2_C, based on the XRD data.

The use of the AHM precursor (Figure 10a) did not lead to the formation of molybdenum nitride; only traces of tetragonal β-Mo_2_N modification, the cubic phase of metallic molybdenum, and monoclinic MoO_2_ started to occur at 700 °C. Higher temperatures resulted in metal molybdenum and MoO_2_. Nitridation of HMT-AHM (Figure 10b) gave a cubic modification of γ-Mo_2_N accompanied by a small α-Mo_2_C fraction at 700 °C. Formation of β-Mo_2_N was observed at 800 °C and only metallic molybdenum was formed at 900 °C. The sample prepared from the HMT+AHM-S (2:1) precursor at 700 °C was cubic Mo_3_N_2_ with a small amount of α-Mo_2_C. The other precursors examined by XRD (Figure 11) produced a mixture of β-Mo_2_N with MoO_2_, in the case of the precursor HMT+AHM-S (1:1) (the mixture with a small proportion of metal molybdenum). An exception was for the previously mentioned HMT+AHM-S (2:1), where cubic Mo_3_N_2_ was formed with a small proportion of α-Mo_2_C. The MoO_3_ precursor was only partially reduced to MoO_2_ and produced a very small proportion of metallic molybdenum.

The crystallite sizes of the nitride phases are shown in Table 2. The biggest crystallite size was related to the β-Mo_2_N phase and determined at 37.4° 2 theta. The value was in two ranges of about 18 and 27 nm. Crystallite sizes of other presented phases γ-Mo_2_N (13.8 nm) and β-Mo_2_N (11.7 nm) were calculated at 37.5° and 37.7° 2 theta, respectively.

Optical and electron micrographs did not show any noticeable difference between molybdenum carbide and nitride samples (Figure 12 and Figure 13). Nitridation of HMT-AHM and HMT+AHM-S (2:1) precursor complexes also resulted in a very crystalline product, while HMT+AHM-M produced a microcrystalline product (Figure 12). The sponge structure was also typical for these materials (Figure 13), but was more distinct and produced labyrinths in comparison to carbides, where relatively small isolated circular pores were observed. When the HMT+AHM-S (1:1) precursor was used, a mixture of two different phases was observed (Figure 14a). A crystalline phase was composed of tugarinovite (MoO_2_) and sponge phases consisted of β-Mo_2_N phases. The same crystalline phase was observed when using the MoO_3_ precursor (Figure 14b).

The obtained data show that the investigated precursors had a high influence on the physico-chemical properties of the prepared materials. It was found that the carbide phase was easier to obtain than the pure nitride phase, which was observed only when using the HMT-AHM precursor at 800 °C, while the carbide phase was present at a temperature above 600 °C for all used precursors except AHM, giving mainly MoO_2_. The summarized scheme of the carburization/nitridation process of HMT-AHM is presented on Figure 15.

Another fact, confirming that the carbides were produced more easily (compare to nitrides), was evidenced by the presence of α-Mo_2_C formed from HMT-AHM and HMT+AHM-S (2:1) at 700 °C as a concomitant phase. Basing on these results, we can conclude that HMT-AHM and HMT+AHM-S (2:1) were the most effective precursors of molybdenum carbides and nitrides synthesis. Even though they were very similar, they were not identical materials.

### 3.4. Supported Samples

Basing on the synthesis of the non-supported molybdenum carbides and nitrides, the HMT-AHM precursor was chosen to prepare supported samples. HMT+AHM-S (2:1) on Al_2_O_3_^#^ was used for the comparative study. The preparation temperature was 700 °C as a compromise between the formation of the desired phase and the avoidance of structural changes in the supports. The samples exhibited high molybdenum contents ranging from 22.1 to 38.4 wt% (Table 3). As a consequence, the high content of the carbide (MoC_x_) or nitride (MoN_x_) phases decreased the initial specific surface area of the SBA-15 and BEA supports.

When using the AZF support, the main purpose was to achieve a higher possible molybdenum loading to get a composite material with large cavities (up to tens of micrometres) filled with molybdenum carbide or nitride crystal particles. This was reasoned by the fact that the larger particles of carbide and nitride phases have better stability during the reactions and increased resistance to complete oxidation to crystalline metal oxides [49,50]. These materials were characterized by having the lowest specific surface area (16–28 m^2^/g), determined mainly by the surface of the bulk nitride or carbide phases located in macroporous cavities of the AZF support. The significant decrease of the area from the original was due to clogging micropores, as is the case of the BEA samples. However, there was no noticeable difference in comparing nitride and carbide samples in S_BET_ reduction. Both groups showed very similar surface area values without any prevalence in blocking pores (Table 3).

The carbides except supported on TiO_2_ (0.16% of N) contained no nitrogen, or below the detection limit of the device. The carbon content varied between 0.49 and 2.57% for all samples. A small amount of carbon (0.08–0.45%) was inherent for each nitride sample and nitrogen was in similar values (0.21–2.38%) as carbon in carbides (Table 3). The low nitrogen and carbon contents could be caused by low amounts of impregnating complexes on ZrO_2_ and TiO_2_, which in turn was due to their low pore volumes.

XRD results (Figure 16) showed that when Al_2_O_3_ was used as the support, well-dispersed carbides and nitrides phases were formed. In carbides samples, apart from α-Mo_2_C and Al_2_O_3_, the main phase was detected as amorphous. The nitrides, besides the main amorphous phase, also contained β-Mo_2_N accompanied by cubic Mo_3_N_2_ and Al_2_O_3_. In addition, the formation of a certain proportion of a cubic AlN phase cannot be excluded. The phase composition of MoC_x_ on Al_2_O_3_^#^ and Al_2_O_3_ was identical, but in the case of MoN_x_, the samples differed by predominant monoclinic MoO_2_ on Al_2_O_3_^#^. The synthesis of Al_2_O_3_^#^ supported materials is more often used than the method with precursor complexes due to its simplicity [51,52,53]. However, in the case of nitrides, there was no complete conversion to the desired phase in these conditions. TiO_2_, in the case of carbide preparation, comprised the crystalline phase of anatase accompanied by the well-dispersed carbide phase containing α-Mo_2_C and β-Mo_2_C in the ratio of 40:60. The possible formation of a small fraction of the cubic delta phase cannot be excluded. The nitride pathway led the β-Mo_2_N phase accompanied by cubic Mo_3_N_2_. The dominant crystalline phase was anatase. A shift of the peak at about 38° 2 theta to the right can be explained as a partial overlapping of the carbide/nitride peak with the anatase phase of the support. A similar composition of the active phases was observed on ZrO_2_, which due to the small pore volume, contained only about half of the amount of molybdenum against the TiO_2_ support. The proportion of the carbide phases was identical, whereas, for nitrides, it was not possible to exclude the presence of the gamma phase. At the carbide pathway, the SBA-15 mesoporous silica provided a mixture of α-Mo_2_C and β-Mo_2_C at a ratio of about 40:60; in the case of the nitride pathway, there was a cubic Mo_3_N_4_ and a part of the monoclinic MoO_2_. The diffractogram, when measuring low angles, shows that the mesoporous structure was preserved (Figure 16). The obtained diffractograms also show the presence of reflections characteristic of microporous zeolite Beta and well-dispersed β-Mo_2_C and cubic Mo_3_N_2_ of carbide and nitride samples, respectively. Diffractograms of the samples prepared on AZF support show the presence of the zeolite phase (clinoptilolite) and crystalline α-Mo_2_C and β-Mo_2_C (approximately 60:40) and β-Mo_2_N, respectively. The influence of the support acidity on the formation of carbide or nitride phases was not clear. On the more acidic supports, in the case of carbides, a mixture of α- and β- phases was produced, while in the mixed nitride phase, the crystalline cubic phase of Mo_3_N_2_ was observed.

The microphotographs from the optical and electron microscope (Figure 17 and Figure 18) show the structure of the prepared supported samples. The composite samples on AZF had macroporous cavities filled with crystalline carbides (Figure 17). The porous structure did not change, even after heat treatment at 700 °C. On the detailed SEM image, a sponge-like structure with a large number of small pores is clearly visible. The samples prepared on the SBA-15 support are shown in Figure 18. A regular structure of the mesoporous silica SBA-15 was not disordered during the preparation of the carbides or nitrides, even at 700 °C. Its surface was formed by nitride or carbide crystals, probably shaped by precursor leakage from the cylindrical pores during heat treatment. The TEM image of carbided SBA-15 (Figure 18b) clearly shows in detail the basic mesoporous structure of SBA-15 that contained the MoC_x_ nanocrystals inside the cylindrical pores and outside on the catalyst surface. Moreover, the dramatic decrease in the surface area of SBA-15 was confirmed by TEM, as the blockage caused by the formation of crystals inside the long cylindrical mesoporous structure of the support (Figure 18b).

The non-supported nitride sample was applied by Murzin’s group in the direct hydrodeoxygenation of algal lipids extracted from *Chlorella alga*, and found to be a perspective catalyst [54]. Other samples have been tested in different hydrotreatment reactions under batch and flow conditions conducted at UniCRE. Further evaluation of the results are under the publishing process.

According to the literature [22,35,36,44,45,55,56,57,58,59,60,61,62,63,64,65,66,67,68,69], the main precursor for the synthesis of supported molybdenum catalysts with the nitride or carbide active phases is considered to be AHM (MoO_3_), and less often the mixture of HMT with AHM, while the preparation of supported catalysts by the impregnation of the HMT-AHM complex is not practically used. AHM impregnated supports are initially calcined in the air to form MoO_3_ on the surface of the support. The most commonly used one is Al_2_O_3_ followed by SiO_2_/SBA-15. In the case of Mo_2_C containing catalysts, carbon (activated carbon, carbon black, nanotubes, etc.) is preferably used, which contributes to the formation of the carbide phase on the support surface.

## 4. Conclusions

This study set out the possibility of molybdenum carbide and nitride synthesis using various precursors and reaction conditions. Based on the analysis of the studied supported and bulk materials, the most suitable preparation conditions to obtain the desired phases were considered. The ability to prepare carbide and nitride phases was demonstrated on the commonly used supports Al_2_O_3_, TiO_2_, ZrO_2_, SBA-15, and zeolite Beta, and also on the less common AZF support. It was investigated that the synthesis of target molybdenum nitrides strongly depends on the structure of the precursor and temperature conditions, while the synthesis of carbide samples always led to the target phase composition. Unlike the carbide samples, where the α-Mo_2_C phase was predominant during nitridation, the mixture of β-Mo_2_N and MoO_2_ with a small amount of metal molybdenum was generally formed. However, using the precursor complex obtained from the mixture of hexamethylenetetramine with ammonium heptamolybdate (HMT-AHM), the pure phase of molybdenum nitride was achieved at 800 °C. At 700 °C, γ-Mo_2_N with a small amount of α-Mo_2_C was formed from the same precursor. A similar situation occurred when using the precursor synthesized by evaporation of the ammonia solution of HMT with AHM at a molar ratio of 2:1, where the resulting cubic Mo_3_N_2_ phase was also accompanied by a small amount of α-Mo_2_C at 700 °C. Supported samples, even at the high molybdenum content, had the high dispersion of the phases. All carbide samples were composed of α- and β-Mo_2_C mixtures. Nitrides supported on Al_2_O_3_ consisted of β-Mo_2_N, and on TiO_2_ and ZrO_2_ consisted of β- and γ- Mo_2_N mixtures, SBA-15, and BEA, despite the β- and γ-phases also containing Mo_3_N_2_ and Mo_3_N_4_. The composite samples prepared on the foamed AZF support included cavities filled with crystalline α- and β-Mo_2_C in the case of carbides and the β-Mo_2_N crystalline phase for nitrides.

## Figures and Tables

**Figure 1 materials-12-00415-f001:**
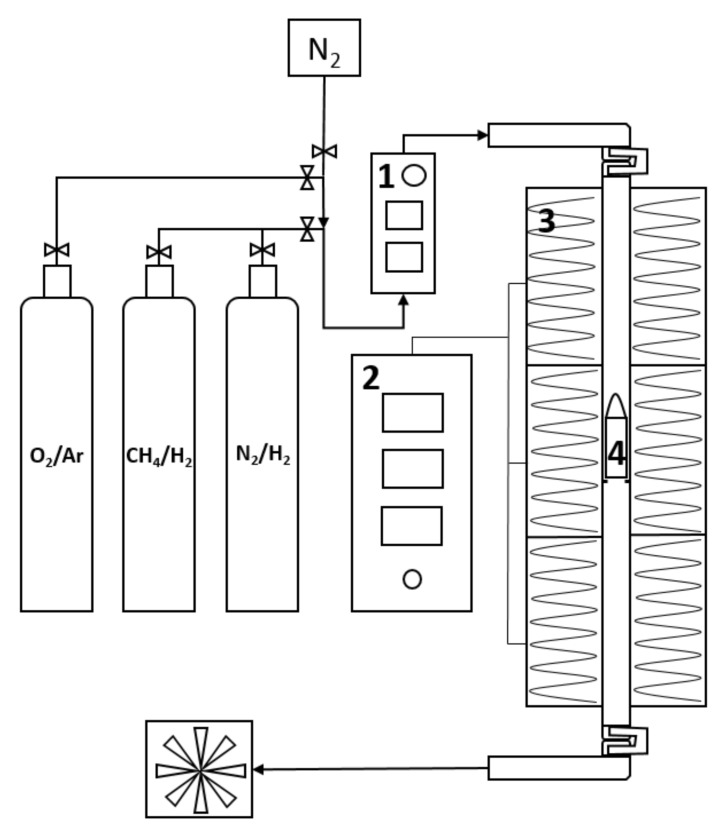
A scheme of the reactor for the preparation of nitrides and carbides, where 1 = gas flow rate controller, 2 = temperature controller, 3 = triple-zone electric oven, 4 = quartz cuvette with a sample.

**Figure 2 materials-12-00415-f002:**
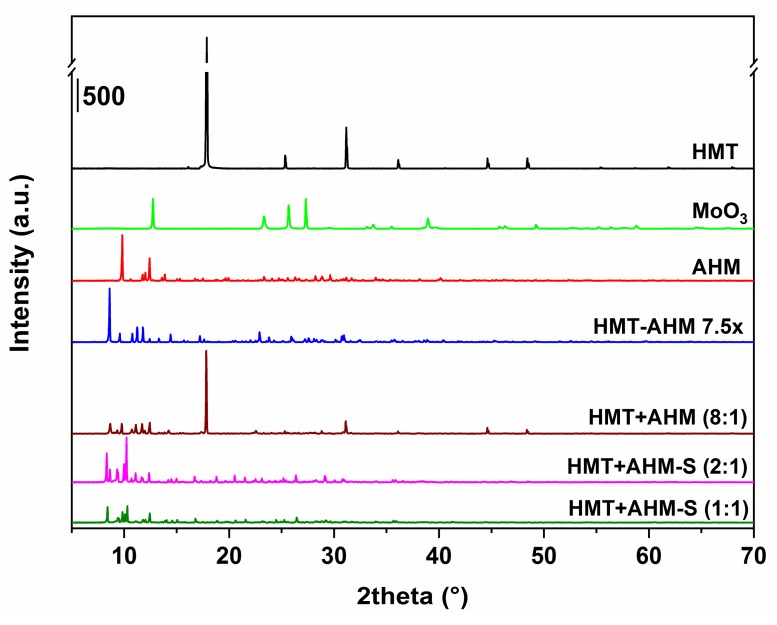
X-ray diffraction patterns of the synthesized precursors and the starting feedstock.

**Figure 3 materials-12-00415-f003:**
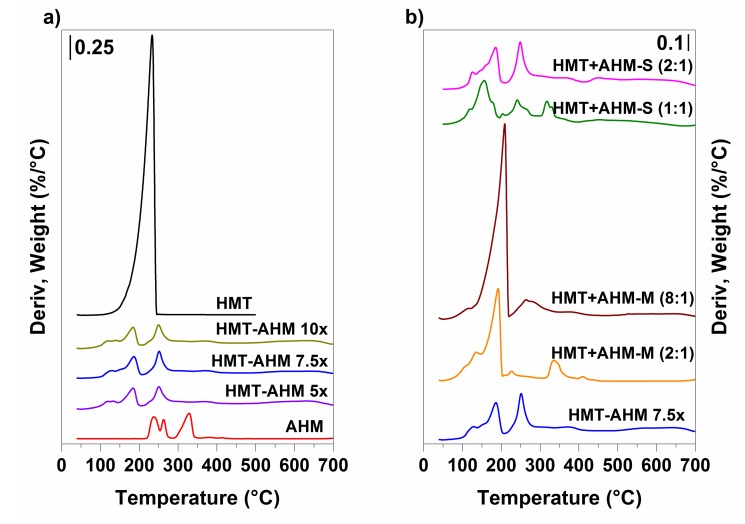
Thermogravimetric analysis of the synthesised precursors (**a**) and the starting feedstock (**b**).

**Figure 4 materials-12-00415-f004:**
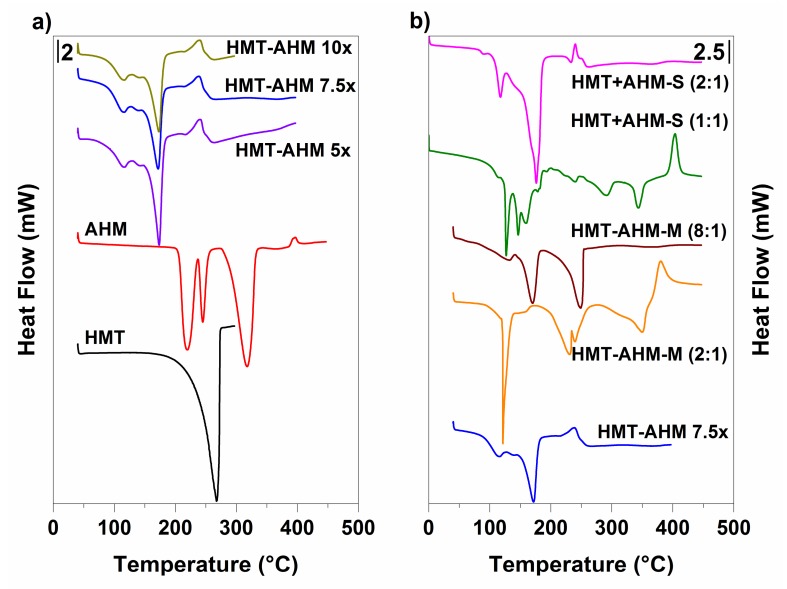
Thermal behavior of the synthesised precursors (**a**) and the starting feedstock (**b**) provided by differential scanning calorimetry (DSC).

**Figure 5 materials-12-00415-f005:**
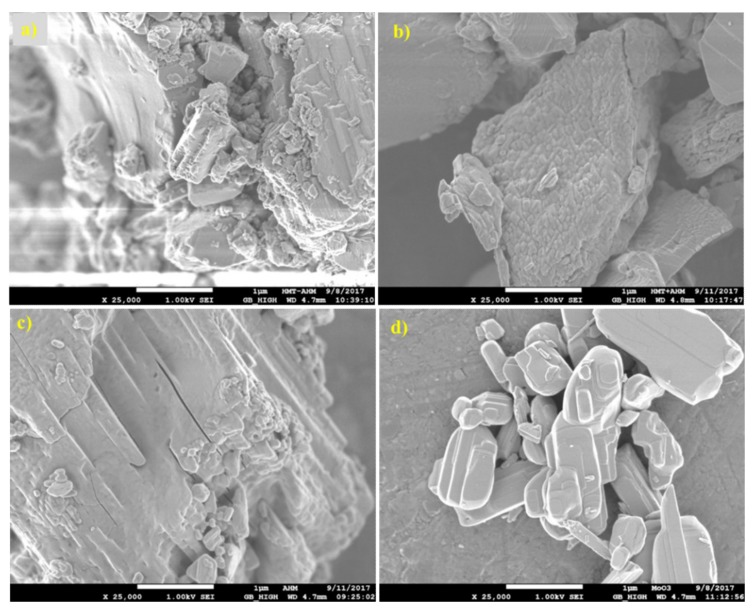
Scanning electron microscopic (SEM) micrographs of the synthesised precursors (**a**) HMT-AHM 7.5× and (**b**) HMT-AHM-S (2:1), and the starting feedstock (**c**) AHM and (**d**) MoO_3_.

**Figure 6 materials-12-00415-f006:**
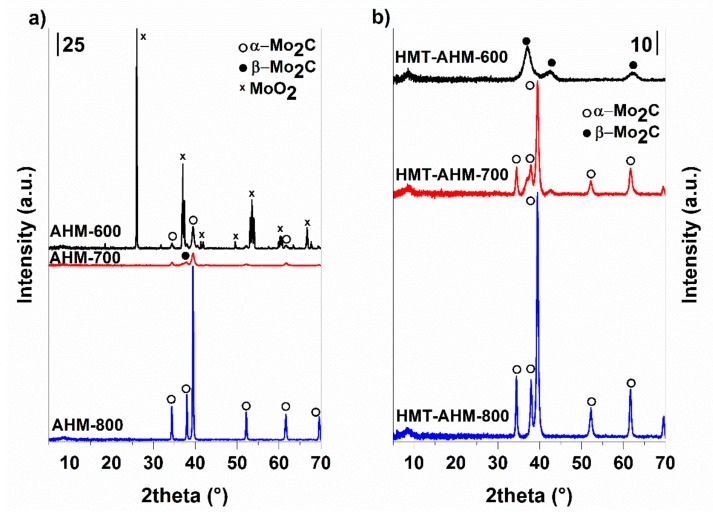
X-ray diffraction patterns of the molybdenum carbides prepared from AHM (**a**) and HMT-AHM (**b**) at 600–800 °C.

**Figure 7 materials-12-00415-f007:**
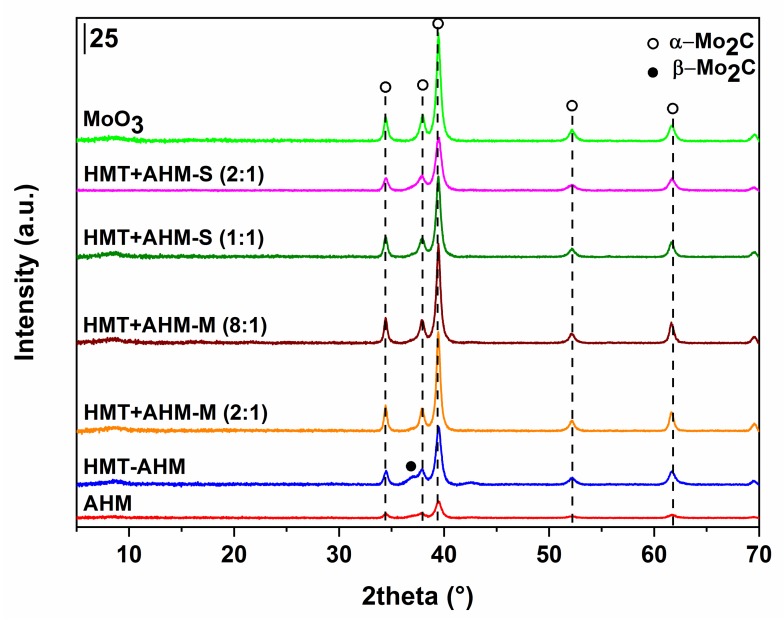
X-ray diffraction patterns of the molybdenum carbides prepared from different precursors at 700 °C.

**Figure 8 materials-12-00415-f008:**
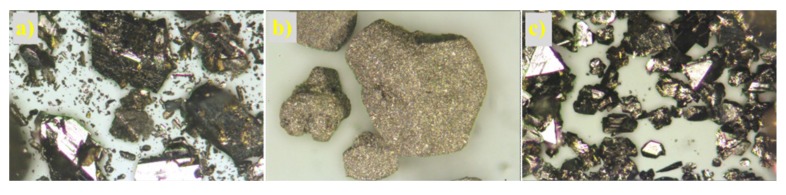
Microscope images of molybdenum carbides at 25× magnification obtained from (**a**) HMT-AHM-700, (**b**) AHM-800, (**c**) HMT+AHM-S (2:1)-700.

**Figure 9 materials-12-00415-f009:**
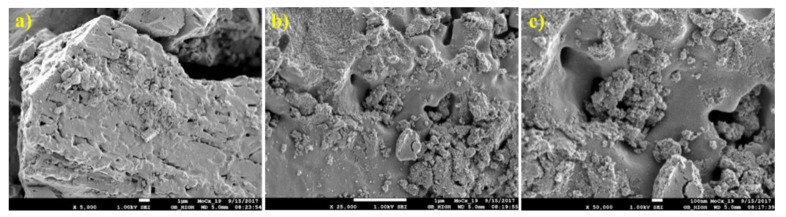
SEM images of molybdenum carbides synthesised from HMT-AHM-600 at magnifications of (**a**) 5000×, (**b**) 25,000×, and (**c**) 100,000×.

**Figure 10 materials-12-00415-f010:**
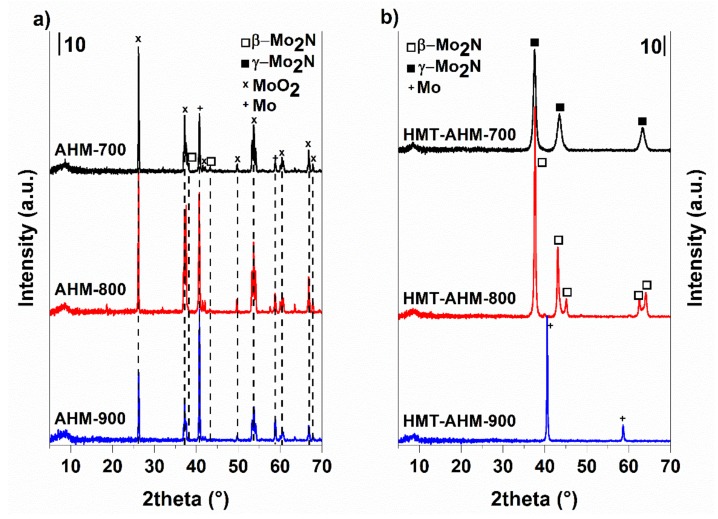
X-ray diffraction patterns of the molybdenum nitrides prepared from AHM (**a**) and HMT-AHM (**b**) at 700–900 °C.

**Figure 11 materials-12-00415-f011:**
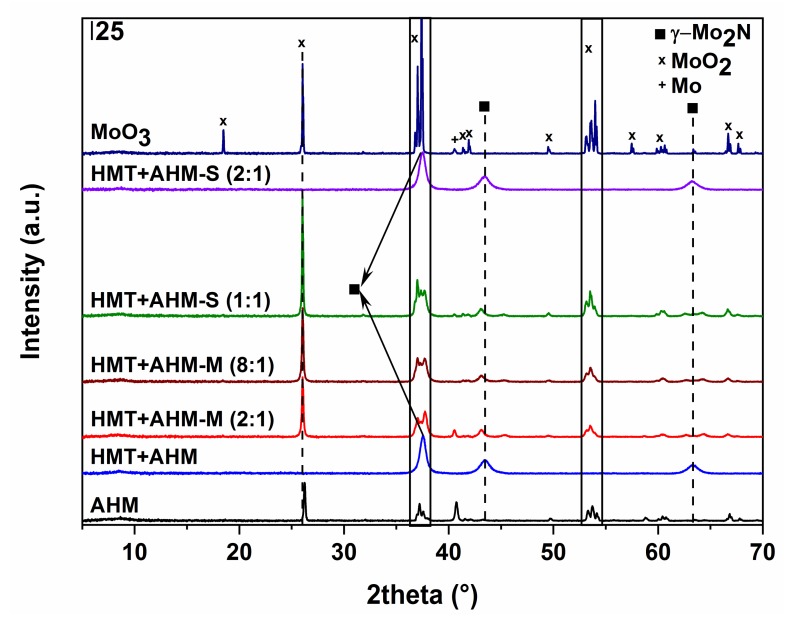
X-ray diffraction patterns of the molybdenum nitrides prepared at 700 °C.

**Figure 12 materials-12-00415-f012:**
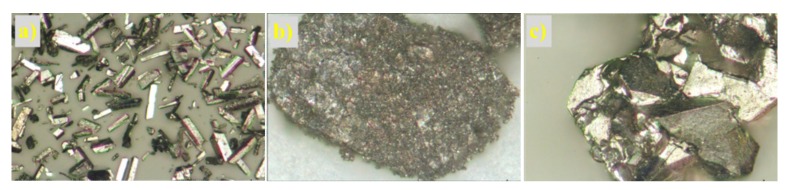
Microphotographs of the samples at 25× magnification: (**a**) HMT-AHM-700, (**b**) HMT+AHM-M (8:1), and (**c**) HMT+AHM-S (2:1).

**Figure 13 materials-12-00415-f013:**
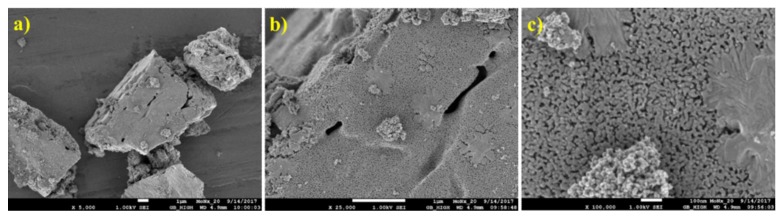
SEM images of HMT-AHM-700 in different magnifications: (**a**) 5000×, (**b**) 25,000×, and (**c**) 100,000×.

**Figure 14 materials-12-00415-f014:**
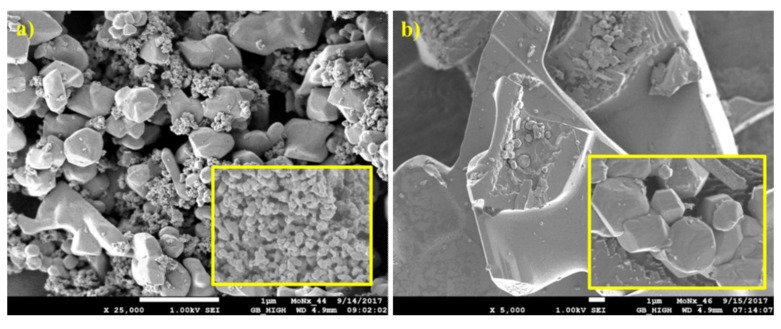
SEM images of (**a**) HMT+AHM-S (1:1) and (**b**) MoO3 prepared at 700 °C (magnifications of 25,000× and 100,000×).

**Figure 15 materials-12-00415-f015:**
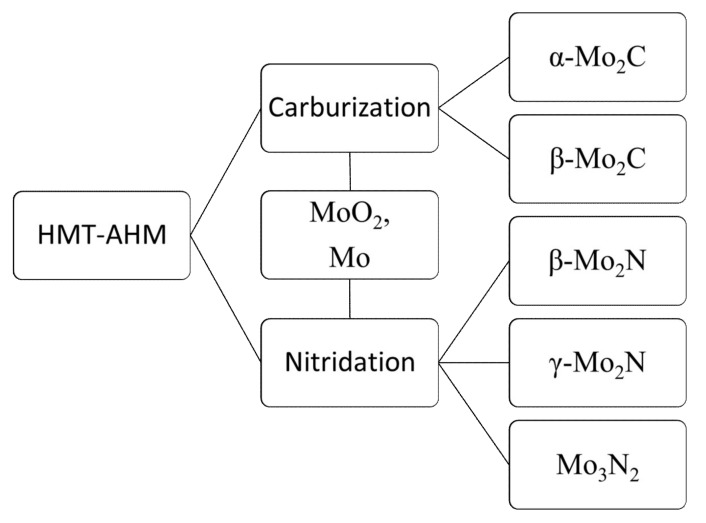
A general scheme of the carburization/nitridation process of HMT-AHM.

**Figure 16 materials-12-00415-f016:**
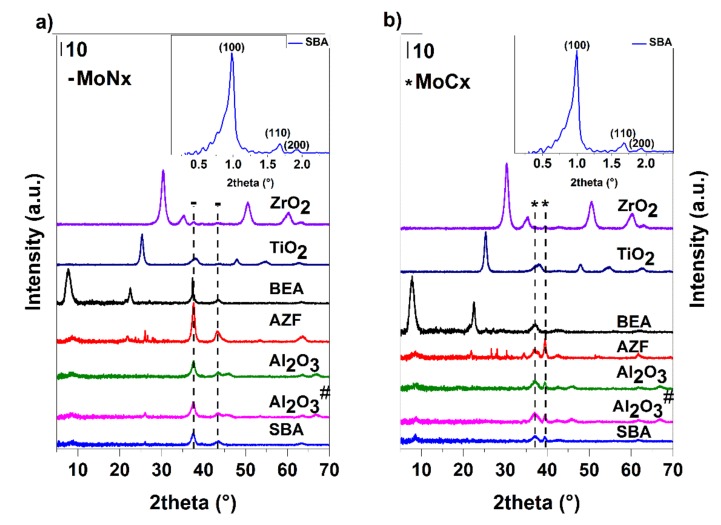
X-ray diffraction patterns of the supported catalysts prepared by different pathways: nitrides (**a**) and carbides (**b**).

**Figure 17 materials-12-00415-f017:**
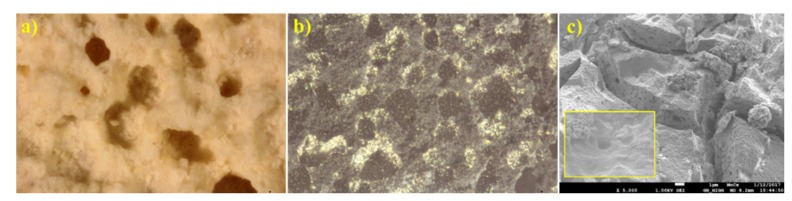
Microphotographs of the samples at 6.3× magnification for (**a**) foamed AZF support, (**b**) carbided AZF composite, and (**c**) detailed structure of the carbide layer made by SEM (5000× and 100,000×).

**Figure 18 materials-12-00415-f018:**
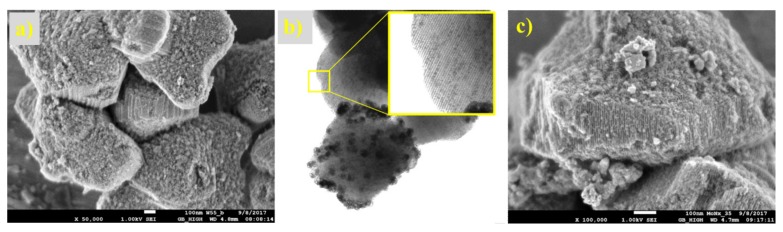
Microphotographs of the samples supported on SBA-15 for (**a**) SEM of the carbide sample (50,000×), (**b**) TEM of the carbide sample, and (**c**) SEM of the nitride sample (100,000×)

**Table 1 materials-12-00415-t001:** Physicochemical properties of the prepared molybdenum carbides.

Sample	S_BET_ (m^2^/g)	Elemental Analysis (wt%)		Content of Crystalline Phase (%)/Crystallite Size D (nm)		
		C	N	β-Mo_2_C	α-Mo_2_C	MoO_2_
AHM-600	3.9	1.64	<0.05	-	12/18.2	88/-
AHM-700	7.4	5.41	<0.05	34/-	66/14.3	-
AHM-800	5.3	6.32	<0.05	-	100/40.2	-
HMT-AHM-600 *	19.4	5.49	<0.05	100/4.9	-	-
HMT-AHM-700 *	25.3	6.52	<0.05	32/-	68/14.6	-
HMT-AHM-800 *	14.1	8.34	<0.05	-	100/21.3	-
HMT+AHM-S (1:1) **	16.2	4.96	<0.05	-	100/18.4	-
HMT+AHM-S (2:1) **	51.8	4.43	<0.05	-	100/15.1	-
HMT+AHM-M (2:1) **	8.7	5.66	<0.05	-	100/21.3	-
HMT+AHM-M (8:1) **	18.3	6.27	<0.05	-	100/19.8	-
MoO_3_ **	0.0	5.81	<0.05	-	100/18.5	-

* HMT-AHM 7.5× precursor, ** preparation at 700 °C.

**Table 2 materials-12-00415-t002:** Physicochemical properties of the prepared molybdenum nitrides.

Sample	S_BET_ (m^2^/g)	Elemental Analysis (wt%)		Content of Crystalline Phase (%)/Crystallite Size D (nm)				
		C	N	β-Mo_2_N	γ-Mo_2_N	Mo_3_N_2_	Mo	MoO_2_
AHM-700	0.0	<0.05	0.37	19/27.4	-	-	36/-	45/-
AHM-800	0.0	0.05	<0.05	-	-	-	57/-	43/-
AHM-900	0.1	<0.05	<0.05	-	-	-	69/-	31/-
HMT-AHM-700 *	25.0	0.45	5.68	-	100/13.8	-	-	-
HMT-AHM-800 *	11.5	<0.05	5.96	100/26.0	-	-	-	-
HMT-AHM-900 *	0.0	<0.05	<0.05	-	-	-	100/-	-
HMT+AHM-S (1:1) **	16.2	0.15	2.44	14/18.3	-	-	-	86/-
HMT+AHM-S (2:1) **	13.2	0.54	5.95	-	-	100/11.7	-	-
HMT+AHM-M (2:1) **	12.2	0.15	2.32	78/17.6	-	-	1/-	21/-
HMT+AHM-M (8:1) **	29.0	0.09	0.97	76/17.5	-	-	-	24/-
MoO_3_ **	0.3	<0.05	<0.05	-	-	-	2/-	98/-

* HMT-AHM 7.5× precursor, ** preparation at 700 °C.

**Table 3 materials-12-00415-t003:** Chemical composition and specific surface areas of the supported samples.

Support	Initial S_BET_ (m^2^/g)	Supported Nitrides						Supported Carbides					
		Chemical Composition (wt%)					S_BET_ (m^2^/g)	Chemical Composition (wt%)					S_BET_ (m^2^/g)
		Mo	X *	Si	C	N		Mo	X *	Si	C	N	
Al_2_O_3_	192	22.5	33.9	0.0	0.08	1.03	123.6	22.1	38.3	0.0	1.46	<0.05	118.8
Al_2_O_3_ ^#^		23.2	38.4	0.0	0.19	0.86	120.1	23.0	37.0	0.0	1.48	<0.05	131.5
TiO_2_	167	17.7	43.9	0.0	0.42	0.53	103.7	16.5	44.5	0.0	1.06	0.16	102.1
ZrO_2_	143	8.2	57.8	0.0	0.16	0.21	117.5	7.8	57.5	0.0	0.49	<0.05	107.1
AZF	120	36.1	1.4	22.8	0.22	2.38	28.1	38.4	1.31	20.1	2.57	<0.05	16.8
SBA	743	25.7	0.0	22.6	0.45	1.92	339.0	36.8	0.0	22.6	2.07	<0.05	340.1
BEA	680	28.3	1.9	27.5	0.31	1.47	331.7	24.8	1.81	25.9	1.07	<0.05	330.5

X * = Al (Al_2_O_3_, Al_2_O_3_
^#^, AZF and BEA), Ti (TiO_2_) and Zr (ZrO_2_).

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
