# Peer review of "Key Role of Precursor Nature in Phase Composition of Supported Molybdenum Carbides and Nitrides"

_materials, 2019, doi:10.3390/ma12030415_

Reviewer 1 Report

The present article describes the preparation and characterization of Mo carbides and nitrides. In particular the modification of their structure induced by different supports is described. The materials are well characterized by different techniques including XRD, BET and SEM

I recommend the acceptance of the manuscript after revision

-Figure 2 should be improved

-Particles size of unsupported and supported Mo based materials should be calculated using both XRD and SEM

Author Response

Point 1: Figure 2 should be improved

Response 1: We did not understand this point. Which way the Figure 2 should be improved? If the point is about reference peaks for XRD, they are all presented in Figure 2 as HMT, MoO3 and AHM as the separate pure substances. The HMT-AHM complex is not presented in the latest X-ray PDF 2018+ database. So we compared the synthesised precursors to the starting feedstock, what means HMT, MoO3 and AHM.

Point 2: Particles size of unsupported and supported Mo based materials should be calculated using both XRD and SEM

Response 2: That point is also not clear for us. From our best knowledge, XRD and SEM cannot be used for that purpose. XRD determines the size of crystallites (presented in the manuscript) – not the particle size and by SEM, we are able to analyse the surface of materials. The resulting materials are agglomerates of non-isolated nanoparticles.

Reviewer 2 Report

Authors describe the synthesis of molybdenum -carbides and -nitrides (analyzing the different phases obtained) using different mixtures of identical precursors (ammonium heptamolybdate, hexamethylenetetramine and aqueous ammonia) in different preparation conditions. From the starting conditions and results found without supports, the authors have created the different carbides and nitrides on different supports. The study is interesting but would deserve some slights improvements before publication.
1- In the abstract, the first sentence could be erased since it does not reflect the results.
2- A general scheme with the precursors would be relevant to a better understanding.
3- It is not clear when ammonia is used or not. Indeed, In part 2.2, the precursors 7.5x seemed to have been treated with ammonia but it is not written (as it should be l. 115)
One comment that was not pointed out by the authors is that the ammonia solution with heptamolybdate might drastically change the nature of the starting molybdenum precursors. POMs are known to be strongly pH sensitive and basic conditions might modify the structure.

4- According to what is presented in the present paper, authors should not call the formed species "catalysts" because no catalytic activity has been presented.

Those minor modifications are necessary for a better understanding.

Author Response

Point 1: In the abstract, the first sentence could be erased since it does not reflect the results.

 Response 1: Corrected.

Point 2: A general scheme with the precursors would be relevant to a better understanding.

 Response 2: The scheme was added to the manuscript.

Point 3: It is not clear when ammonia is used or not. Indeed, In part 2.2, the precursors 7.5x seemed to have been treated with ammonia but it is not written (as it should be l. 115)

One comment that was not pointed out by the authors is that the ammonia solution with heptamolybdate might drastically change the nature of the starting molybdenum precursors. POMs are known to be strongly pH sensitive and basic conditions might modify the structure

 Response 3: Corrected. We did a mistake in Line 114. The initial precursor HMT-AHM was synthesised without ammonia using the method reported by Afanasiev. Ammonia was used only for the samples HMT+AHM-S as written in 123-124 and for the sample supported on Al2O3#.

 In this work, we did not touch on the topic of the influence of the ammonia solution on HMT structure and the pH sensitivity of POMs in general. The article is primarily about the properties and use of the HMT-AHM complex for the preparation of nitrides and carbides, where NH3 was used only as a solvent for impregnation of the supports. The goal was to get a high Mo loading on the support. The NH3 solution was applied also due to the fact that the HMT-AHM complex has low solubility in water. We did not monitor any changes because it was not an object. But we are thankful for this comment, so we will carefully analyse the effect of pH conditions in future studies.

Point 4: According to what is presented in the present paper, authors should not call the formed species "catalysts" because no catalytic activity has been presented.

 Response 4: Corrected

Reviewer 3 Report

This manuscript describes the use of two different molybdenum complex as precursors for production of molybdenum carbides and nitrides. Unfortunately it is not new to apply hexamethylenetetramine in the production of carbide with ammonium heptamolybdate. The novelty of this work is therefore questionable. 

The authors observed some difference between two precursors but did not explain the impact on the final carbide/nitride structure in a detailed way. Currently, this work is not appropriate for publication. Besides this general comment, here are some specific comments for the authors to consider:

1.    Line 187-189, authors claimed that their results were different from the cited reference work but did not explain what was the cause for this discrepancy. 

2.    Figure 2, authors should add reference peaks for XRD.

3.    For elemental analysis, I suggest authors to use atomic% instead of wt%. The previous would give better idea about composition of the carbide.

4.    It will be better if authors could also attach the pore structure information.

5.    Line 345, authors claim that carbides were easier to be produced than nitride. How did authors conclude this?

6.    In Table 3, why did authors use different loading of Mo for impregnation? This introduced another variation that may affect the result. I’d suggest to keep it constant.

7.    Figure 15, the peaks over TiO2 supported sample were shifted from reference, what’s the cause for this?

8.    Figure 17b, a clearer TEM picture should be used. It is difficult to see the channel from the current picture. The authors also claim that there were nanocrystals inside the pores. I do not see this from Figure 17b; the authors might want to add additional evidence to support this claim.

9.    I did not understand why the authors put Table 4 in the manuscript. Authors claimed that HMT+AHM combination is less practically used for impregnation. Then what’s the purpose of this study? Is there any advantage of using this combination instead of other material?

Author Response

“This manuscript describes the use of two different molybdenum complex as precursors for production of molybdenum carbides and nitrides. Unfortunately it is not new to apply hexamethylenetetramine in the production of carbide with ammonium heptamolybdate. The novelty of this work is therefore questionable.

The authors observed some difference between two precursors but did not explain the impact on the final carbide/nitride structure in a detailed way.”

The purpose of the work was not to show the use of two different complexes, but the comparison of the synthesis of molybdenum carbides and nitrides using different precursors (AHM, HMT-AHM, HMT + AHM, MoO3), and the use of the complex for the preparation of supported catalysts. From our best knowledge, no one has used the complex for impregnation of the support (already patented by us). Further comments are in the Responses

Point 1: Line 187-189, authors claimed that their results were different from the cited reference work but did not explain what was the cause for this discrepancy.

Response 1: Possibly confused due to a mistake in Line 187 HMT+AHM-S. Corrected

We did not study what was the cause for this difference that much in detail.

By X-ray of the monocrystals (HMT-AHM complex and crystals formed by crystallization of HMT with AHM from the NH3 solution), more information could be shown. We did not provide the preparation of the monocrystals, and, from our knowledge, it is not so easy and it was not an object of this work. However, it can be a topic of further study.

The difference in structure may be reasoned be the partial incorporation of ammonia or its hydrates (NH3xH2O) into the crystal structure and/or the change of POMs in the alkaline condition as described by the Reviewer 2.

Point 2: Figure 2, authors should add reference peaks for XRD.

Response 2: The reference peaks are presented in Figure 2 as HMT, MoO3 and AHM as the separate pure substances. The HMT-AHM complex is not presented in the latest X-ray PDF 2018+ database. So we compared the synthesised precursors to the starting feedstock, what means HMT, MoO3 and AHM.

Point 3: For elemental analysis, I suggest authors to use atomic% instead of wt%. The previous would give better idea about composition of the carbide.

Response 3: For that purpose, we would have to measure not only carbon and nitrogen content but also molybdenum with possibly contained oxygen for unsupported samples (Table 1 and 2), which we did not do. The recalculation could be performed only for the supported catalysts (Table 3) where all elements were determined (assuming that oxygen will be residual to 100%), but it would be strange to have in the article one table to indicating the wt% and the other at%.

Point 4: It will be better if authors could also attach the pore structure information.

Response 4: We agree with this point, but measurement and evaluation cannot be provided within 10 days that the Journal gave us for revision. Moreover, due to the already large amount of data, the article will acquire another 2-4 graphs (depending on whether the non-impregnated supports would be mentioned).

Point 5: Line 345, authors claim that carbides were easier to be produced than nitride. How did authors conclude this?

Response 5: From the experimental data described in the chapter above (3.2. Non-supported MoCx): pure carbides were formed from all precursors, even at lower temperatures, and they did not contain other non-carbide phases (MoO2, Mo, etc.)

Point 6: In Table 3, why did authors use different loading of Mo for impregnation? This introduced another variation that may affect the result. I’d suggest to keep it constant.

Response 6: We tried to impregnate the maximum amount of Mo that can be loaded to the supports limited by their absorbency. The goal was to determine the development of the nitride and carbide phases with the maximum content of Mo. The AZF support was repeatedly impregnated to fill the macro-cavities with the HMT-AHM precursor (Afanasiev complex), which would form a bulk layer of nitride/carbide on the walls.

Point 7: Figure 15, the peaks over TiO2 supported sample were shifted from reference, what’s the cause for this?

Response 7: The presented nitride and carbide phases have a low intensities, and the peaks are not as sharp and partially overlap the peak of the anatase phase of the support at about 38° 2 theta, which looks like the shift of the peak to the right.

Clarification was added to the text.

Point 8: Figure 17b, a clearer TEM picture should be used. It is difficult to see the channel from the current picture. The authors also claim that there were nanocrystals inside the pores. I do not see this from Figure 17b; the authors might want to add additional evidence to support this claim.

Response 8: Figure 17b was changed to make it better to see what is inside the pores.

Point 9: I did not understand why the authors put Table 4 in the manuscript. Authors claimed that HMT+AHM combination is less practically used for impregnation. Then what’s the purpose of this study? Is there any advantage of using this combination instead of other material?

Response 9: Table 4 was removed.

The purpose of this work was to show the advantages of the HMT-AHM complex compared to other precursors (MoO3, AHM, HMT + AHM) in the synthesis of unsupported and/or supported "catalysts". One of the advantages of using the Afanasiev complex is the simpler control of the phase composition and easier preparation of pure nitride/carbide phases. Another advantage is technological, and this means better solubility of the complex in ammonia than the solubility of individual components (HMT and AHM), which is important when impregnating supports. This makes it easier to prepare catalysts and increase the content of the active phase (MoNx/MoCx).

Round  2

Reviewer 1 Report

The manuscript can be accepted after minor revision. The authors should modify the color for HMT+AHM (8:1) which is not clear (Figure 2)

Author Response

Point 1: The authors should modify the color for HMT+AHM (8:1) which is not clear (Figure 2)

Response 1: Figures 2, 3, 4, 7, 11, containing HMT+AHM (8:1), were modified.

Reviewer 3 Report

the authors revised their manuscript which can now be accepted

Author Response

Point 1: The authors revised their manuscript which can now be accepted

Response 1: We appreciate for all the comments that helped us improve the manuscript before its acceptance.
